# Tensile Fracture Behavior and Characterization of Ceramic Matrix Composites

**DOI:** 10.3390/ma12182997

**Published:** 2019-09-16

**Authors:** Jeongguk Kim

**Affiliations:** Advanced Railroad Vehicle Division, Korea Railroad Research Institute, Uiwang 16105, Korea; jkim@krri.re.kr; Tel.: +82-31-460-5518

**Keywords:** ceramic matrix composites, cross-ply, mechanical characterization, high-speed infrared camera, microstructural analysis, failure mode, failure mechanism

## Abstract

Tensile fracture behavior of ceramic matrix composites (CMCs) was investigated using characterization tools. First, a high-speed infrared camera was used to monitor the surface temperature of the CMC specimen during mechanical testing. An infrared camera is a tool used to detect infrared (IR) radiation emitted from a specimen as a function of temperature, and it was used to analyze the temperature monitoring of specimen surface and fracture behavior during the tensile test. After the test, the microstructural analysis using SEM was performed. SEM analysis was performed to investigate the fracture mode and fracture mechanism of CMC materials. In this paper, it was found that the results of the surface temperature monitoring obtained from IR thermal imaging technology and the failure mode analysis obtained through SEM were in a good agreement. These techniques were useful tools to explain the mechanical behavior of ceramic matrix composites. The detailed experiments and testing results will be provided.

## 1. Introduction

Ceramic matrix composites are spotlighted for high temperature structural applications [1,2]. In general, ceramics are useful materials for high temperature materials due to their high strength, high temperature and corrosion resistance. They are also lightweight, but their use is limited because of their unique brittle characteristics. In order to compensate for such brittleness and to increase fracture toughness, various types of fibers could be reinforced to form ceramic composite materials [1,2,3,4].

Various types of ceramic matrix composites can be prepared. For example, ceramic composite materials can be manufactured in various forms, such as particulate reinforced ceramic matrix composites and short or continuous fibers reinforced composites. In general, ceramic composites reinforced with continuous fibers are widely used in practical applications, for example, heat exchangers, high temperature blades, automotive brake discs, jet engines, surface parts of spacecraft, etc. [1].

The main advantages of ceramic matrix composites as compared to monolithic ceramics are shown in Figure 1. In the stress-strain behavior, monolithic ceramics show only elastic regions, and exhibit tensile properties in the form of brittle fracture without any plastic deformation. However, in the case of ceramic matrix composites reinforced with continuous fibers, at the beginning of the uniaxial tensile test, they exhibit an elastic region like monolithic ceramics, and then reach an ultimate tensile strength (UTS) through a proportional limit, followed by the so-called graceful period similar to the plastic deformation of the metallic materials, and subsequently to the final fracture [1,3,5,6,7,8].

Although there is no plastic deformation in ceramic matrix composites in practice, the reason for the similar properties of deformation in the stress–strain behavior is due to the influence of the continuous fibers present in the ceramic matrix materials. In general, the following process is possible; when the matrix is broken first (matrix cracking), then the crack propagates to the inside ceramic matrix, and when the crack tip meets the continuous fibers, debonding between the fiber and the matrix is occurred followed by extensive fiber pullouts. Finally, as the ceramic fibers take time for the final failure and improve the fracture toughness, thereby obtaining a stress–strain behavior similar to the plastic deformation in metallic materials. As a result, in the case of the ceramic composite materials reinforced with continuous fibers, it is observed that the fracture toughness can be improved [1,5,8].

Many studies have been made to observe the fracture behavior of ceramic matrix composites. Various forms of nondestructive evaluation (NDE) techniques are applied to aid this research [9,10]. Techniques such as acoustic emission, high-speed infrared camera (Infrared Thermography), and ultrasonic testing are commonly used, and acoustic emission and high-speed infrared camera are useful for monitoring various signals generated from specimens during mechanical testing [11,12,13,14,15].

In practice, it is possible to apply these techniques to analyze the mechanical properties of materials with signals from nondestructive evaluation techniques. In particular, in the case of a high-speed infrared camera, as a device for detecting infrared, the temperature distribution of the two-dimensional surface of an object can be displayed. Applying this technology to the uniaxial tensile test or fatigue test is able to monitor the temperature change of the material surface in real time. That is, it can be used to analyze the failure mode or failure process by monitoring the temperature change of the material surface during the tensile test.

In this study, a high-speed infrared camera was used for the temperature monitoring of the specimen surface, and to investigate the uniaxial tensile failure characteristics of ceramic matrix composites. Furthermore, the purpose of this study was to analyze the relationship between the temperature changes obtained from infrared camera and mechanical testing data. After the tests, SEM microstructural characterization was performed to analyze the failure mode and failure mechanism of ceramic matrix composites, and to provide a detailed description of the failure process.

Therefore, the main objectives of this study are as follows: (1) to study the uniaxial tensile failure characteristics of ceramic matrix composites; (2) to monitor specimen temperature during tensile testing using infrared thermography technology as a nondestructive evaluation technique; (3) to analyze fracture characteristics of ceramic matrix composites through SEM microstructural characterization; and (4) to be utilized in the material design by providing the fracture characteristics of the ceramic matrix composites.

## 2. Materials and Experimental Methods 

### 2.1. Materials

The materials used in this study are Nicalon ceramic fiber reinforced ceramic matrix composites with calcium aluminosilicate (CAS, CaAl_2_Si_2_O_8_) glass ceramic matrix. Nicalon fiber (Nippon Carbon Co., Ltd., Tokyo, Japan) is a ceramic fiber composed mainly of SiC, and has a diameter of about 10 to 15 µm. Nicalon/CAS composites were manufactured by the hot-pressing method, and [0/90]_4S_ cross-ply composite panel was used in this investigation. The microstructure of the Nicalon/CAS specimen is shown in Figure 2.

### 2.2. Tensile Testing and Temperature Monitoring

The dog-bone type tensile specimens were prepared from [0/90]_4S_ cross-ply Nicalon/CAS panel for the uniaxial tensile test, and the tensile test was performed using a servohydraulic testing machine (MTS 810, MTS, Eden Prairie, USA). Tensile tests were carried out in displacement control mode at room temperature with a cross-head speed of 0.3 mm per minute with the ASTM code [16]. In order to understand the relationship between changes in mechanical properties and temperature during the tensile test, a high-speed infrared camera (A8300, FLIR, Santa Barbara, USA) was used to monitor the entire tensile test. The use of an infrared camera makes it possible to monitor the temperature change of the specimen during the tensile test, and to observe the temperature change of the specimen based on the final fracture point.

In this study, the speed of the high-speed infrared camera was 7 Hz. The testing setup for the tensile test with a high-speed infrared camera is shown in Figure 3. After the tensile test, the fractured specimens were subjected to microstructural analysis using SEM (S-360, Cambridge Instruments, Cambridge, UK). Through the observation of these microstructures, the inferences were made about the correlation between the IR camera monitoring results and the mechanical property data obtained in the tensile test.

## 3. Results and Discussion

As mentioned in Figure 1, there is a difference between monolithic ceramics and ceramic composites for uniaxial tensile testing. In general, monolithic ceramics that are not reinforced with fibers or other particles do not exhibit plastic zones, but show typical brittle fracture behavior leading to fracture immediately after passing through the elastic zones. On the other hand, ceramic matrix composites reinforced with continuous ceramic fibers show the same elastic deformation as monolithic ceramics until the first matrix cracking occurs. Afterwards, it shows the shape of the plastic zone that can be seen in the metallic materials through further processes such as further matrix failure, crack deflection at interface, and more interfacial interactions. This is not a plastic deformation actually occurring, but rather the fracture toughness is increased due to the behavior that increases the time to fracture in the processes of the formation of cracks and fiber pullout, showing the deformation behavior of this type. After continuing to reach UTS, they undergo a process such as fiber pullout, indicating so-called graceful failure, and finally the failure occurs.

Figure 4 shows the tensile test results obtained in this study. As shown in the general tensile fracture behavior of ceramic matrix composites in Figure 1, the stress variation is similar. For both samples (sample 1 and 2), almost similar tensile failure behavior was obtained. However, the final failure occurred immediately after the UTS. Overall, the proportional limit was determined at about 80 MPa and the maximum tensile strengths were about 180 and 195 MPa, respectively.

Figure 5 shows the temperature changes of the cross-ply Nicalon/CAS composites monitored during the tensile test. The characteristics of the temperature changes are that fluctuation or gradual degradation is observed around 0.5 °C, but overall, both samples show no significant temperature change during the tensile test. However, a sharp temperature peak was observed at the time of break, and the instantaneous temperature rise was 2.1 and 4.1 °C, respectively. From these results, no significant temperature change is observed during the tensile test, but it is possible to infer whether it is accompanied by abrupt fracture behavior in the final failure stage, for example, extensive fiber pullout and massive matrix cracking.

Figure 6 combines Figure 4 and Figure 5 to show stress-time and temperature change-time simultaneously. These results can be directly seen by comparing the relationship between stress change and temperature change during the tensile test. The results of the tensile test show that the final fracture occurred instantly after reaching the UTS, which is similar in the temperature change data. That is, a sudden temperature peak is observed just before the final break, and it can be assumed that the final failure was induced by a momentary and continuous fiber breakage. Since the speed of the high-speed infrared camera was 7 Hz, the final breakdown occurred suddenly within 1 second (7 data points representing 1 s). Since one temperature datapoint means about 0.143 s, it can be seen that the mode of final fracture was in the form of fiber pullout and failure in an instant. At this time, it can be seen that the thermal energy is generated due to the pullout energy between the matrix and the fiber and the final fiber breakage, indicating a temperature rise with an instantaneous temperature peak.

Finally, microstructural analysis was performed using SEM to analyze the failure mode or failure mechanism of Nicalon/CAS composites. Figure 7 shows the cross section of the final fracture surface of a Nicalon/CAS composite. Figure 7 shows that the fibers in the 90° laminate are still connected to each other and the fibers in the 0° laminate are completely split up and down.

Therefore, inferring from the results of Figure 7, the 0° fibers composed of the matrix were broken first with matrix cracking and completely separated in the stress loading direction. Subsequently, it can be seen that the 90° fibers withstand the breakage relatively and finally break with the increase in stress. Therefore, the SEM image analysis proved that the final temperature peaks and temperature rises described in Figure 5 and Figure 6 were made by the breakage and pullout of these 90° fibers.

Figure 8 shows the failure image of SEM in 0° laminate of cross-ply Nicalon/CAS composite. It shows the interfacial debonding between fiber and matrix in 0° laminate. Through Figure 8, the initial crack was initiated in a 0° laminate, followed by interfacial debonding and the propagation of the crack by the applied load, and the crack encounters a 90° fiber. It can be inferred that the final breakdown is completed by breaking the fibers in the 90° direction.

Figure 9 shows the final failure SEM images of cross-ply Nicalon/CAS composites with extensive fiber pullout. Figure 9 shows the final cues in more detail. It can be understood that following the break of the 0° laminate, the crack propagates to the 90° laminate so that the final failure is followed by multiple matrix cracking and interfacial debonding around the 90° fiber bundle followed by the final fiber pullout and fiber break. As such, the overall failure mode can be explained. It can also be seen that the temperature peaks and temperature rise immediately before breakdown were due to the final fiber pullout and fiber breakage.

Inferring the above results, it becomes possible to explain the overall failure mode or failure mechanism as follows (1) matrix cracking in 0° laminate, (2) debonding between the fiber and matrix at 0° laminate, (3) further crack propagation in the 0° laminate, (4) delamination between 0° and 90° laminates, (5) fiber debonding and pullout in the 90° laminate, and (6) final rupture with extensive fiber pullout.

## 4. Conclusions

In this paper, the tensile failure characteristics of Nicalon/CAS ceramic matrix composites were analyzed using a nondestructive evaluation technique and SEM microstructural characterization. Tensile tests showed that the overall tensile failure characteristics exceeded the proportional limit and the final fracture occurred in around the UTS. Inferences about these mechanical properties were made through temperature monitoring using a high-speed infrared camera and microstructure observation analysis using SEM. Analysis of the specimen temperature monitoring with a high-speed infrared camera confirmed the fracture process and mode as a function of temperature. It was found that a high-speed infrared camera can be used as a useful tool for inferring failure modes and/or mechanisms. Finally, SEM microstructual analysis was able to confirm the results obtained by tensile tests and temperature monitoring. In conclusion, the tensile failure of Nicalon/CAS ceramic matrix composites was confirmed through the following processes (1) matrix cracking in 0° laminate; (2) debonding between the fiber and matrix at 0° laminate; (3) further crack propagation in the 0° laminate; (4) delamination between 0° and 90° laminates; (5) fiber debonding and pullout in the 90° laminate; and (6) final rupture with extensive fiber pullout.

## Figures and Tables

**Figure 1 materials-12-02997-f001:**
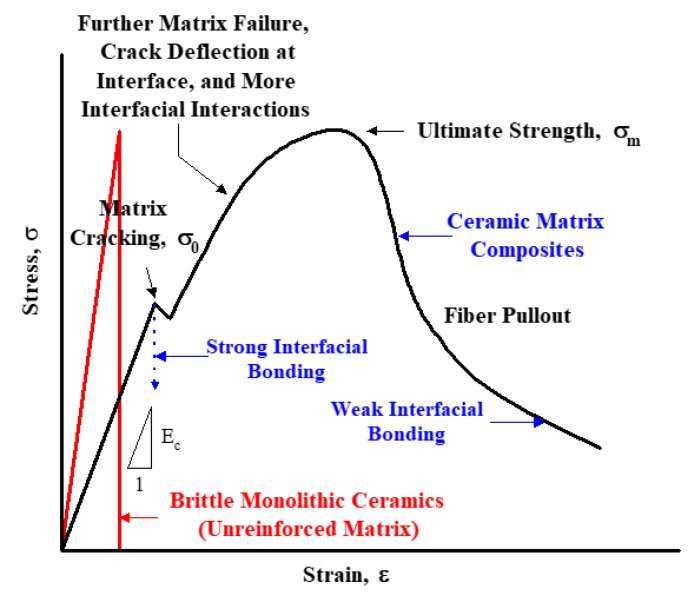
The stress–strain behavior in ceramic matrix composites as compared with monolithic ceramics.

**Figure 2 materials-12-02997-f002:**
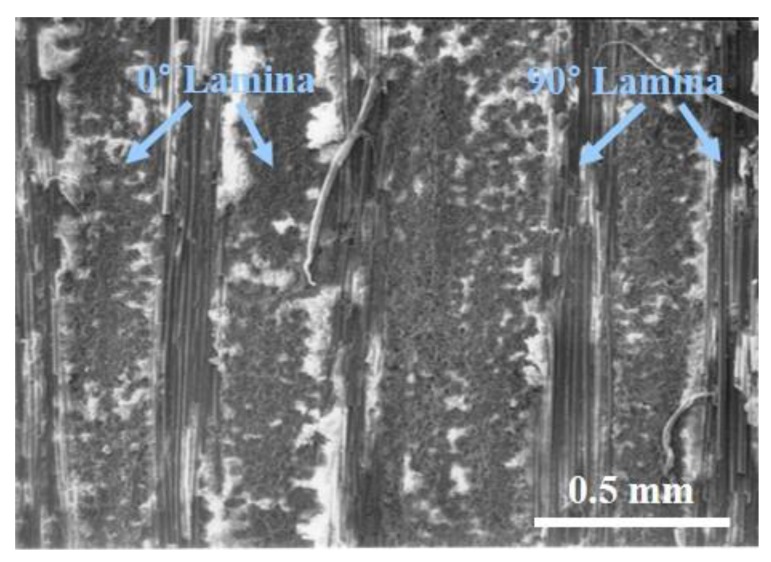
The microstructure of the cross-sectional view of [0/90]_4S_ in Nicalon/CAS composites.

**Figure 3 materials-12-02997-f003:**
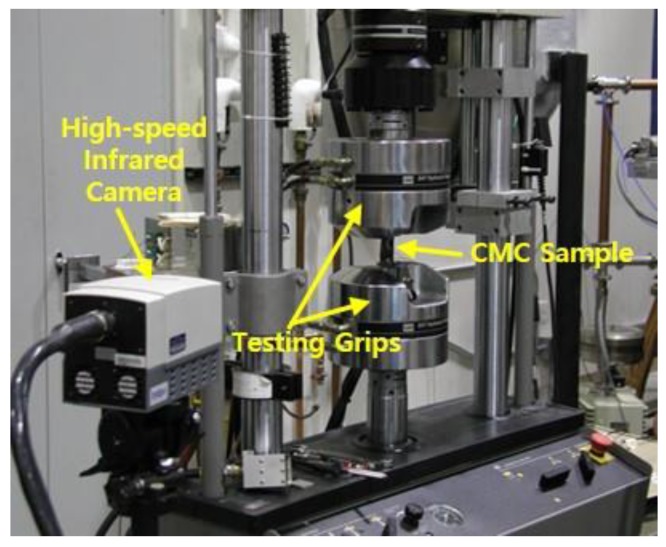
The testing setup for tensile testing with a high-speed infrared camera.

**Figure 4 materials-12-02997-f004:**
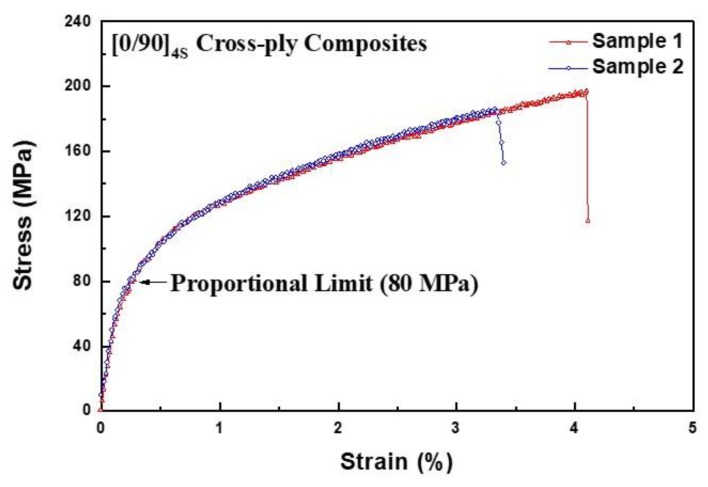
The stress–strain behavior of cross-ply Nicalon/CAS composites.

**Figure 5 materials-12-02997-f005:**
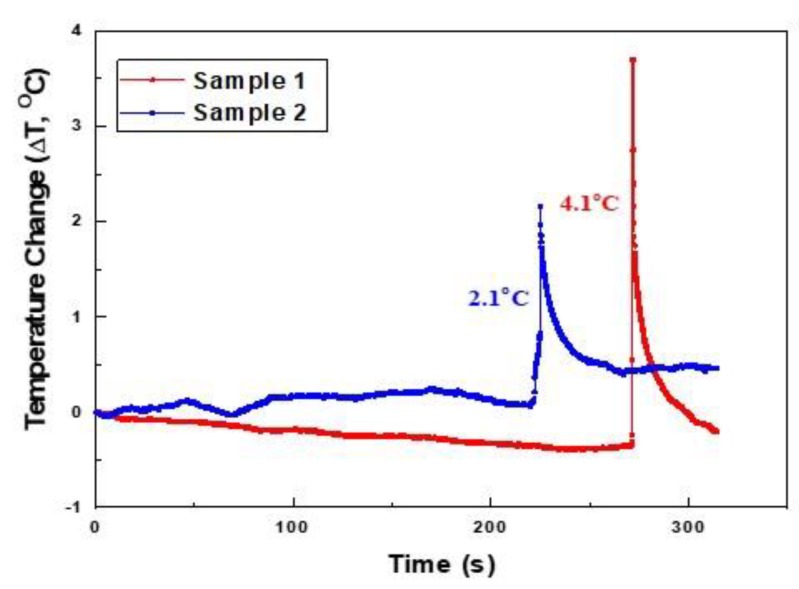
The temperature variation during tensile testing in cross-ply Nicalon/CAS composites.

**Figure 6 materials-12-02997-f006:**
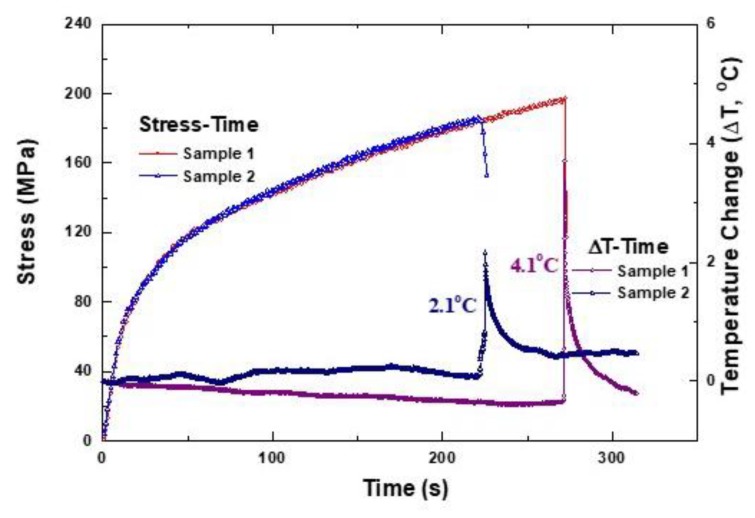
The stress–strain behavior vs. temperature variation during tensile testing in cross-ply Nicalon/CAS composites.

**Figure 7 materials-12-02997-f007:**
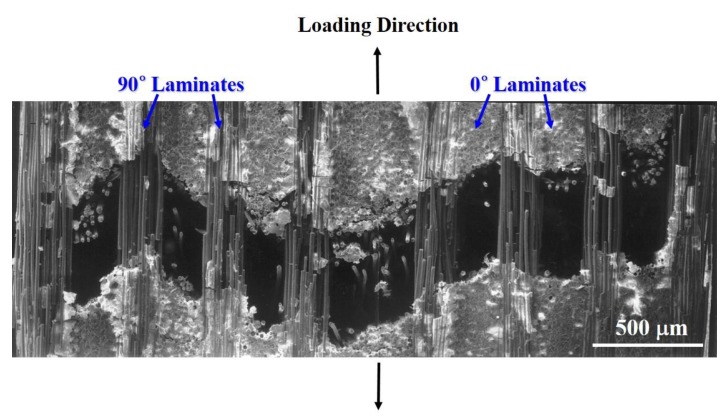
SEM image of the overall fracture cross section in cross-ply Nicalon/CAS composite.

**Figure 8 materials-12-02997-f008:**
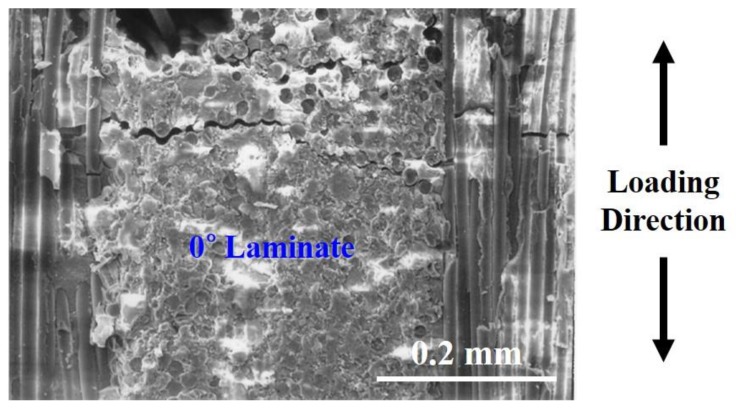
The failure image of SEM in 0° laminate of cross-ply Nicalon/CAS composite.

**Figure 9 materials-12-02997-f009:**
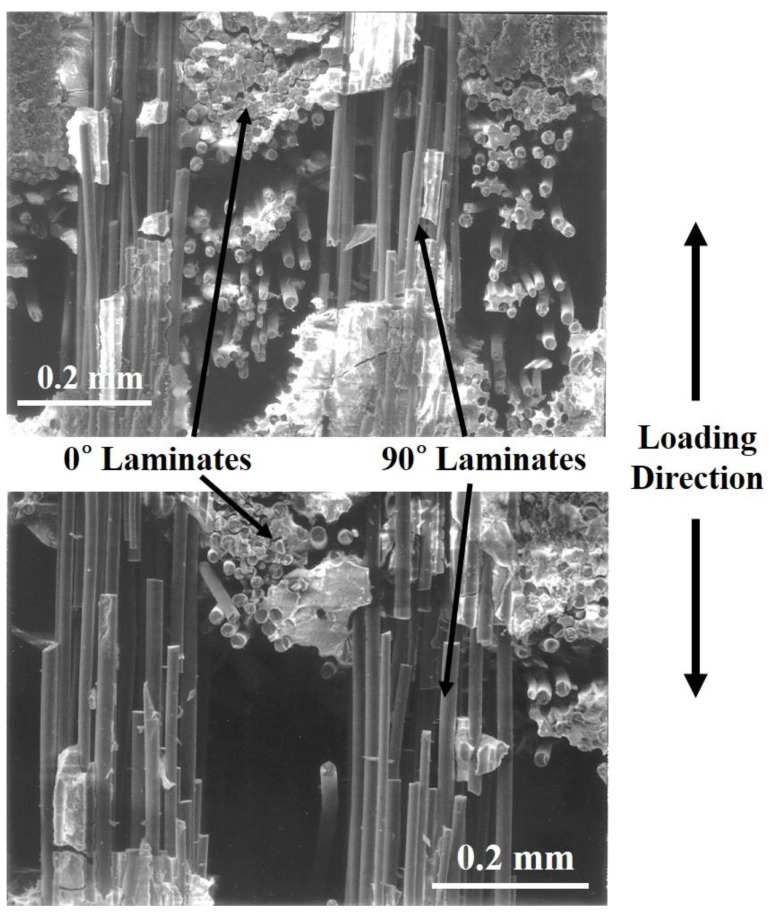
The final failure SEM images of cross-ply Nicalon/CAS composites with extensive fiber pullout.

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
