# Peer review of "Tensile Fracture Behavior and Characterization of Ceramic Matrix Composites"

_materials, 2019, doi:10.3390/ma12182997_

Round 1

Reviewer 1 Report

I suggest indicating the main parts of the testing device in Figure 3.   There is no clear explanation in the text what samples 24-14 and 24-16 means 

Author Response

Response to Reviewer 1 Comments

Point 1: I suggest indicating the main parts of the testing device in Figure 3.

Response 1: In order to indicate the main parts of the testing device, Figure 3 was updated as the reviewer suggested. High-speed infrared camera, testing grips, and CMC sample were added as captions.

Point 2: There is no clear explanation in the text what samples 24-14 and 24-16 means

Response 2: Two types of CMC samples were used in this study, 24-14 and 24-16. As suggested by the reviewers, 24-14 and 24-16 were modified to be Sample 1 and Sample 2 to avoid confusion. Also, they were modified accordingly in Figure 4, Figure 5 and Figure 6.

Reviewer 2 Report

This article entitled “Tensile Fracture Behavior and Characterization of Ceramic Matrix Composites” provides the example of the use of measurements involving an infrared camera and microstructural analysis using SEM pictures to explain the mechanical behaviour of ceramic matrix composites.

During this work, authors were able to confirm that the tensile failure of tested ceramic matrix composites runs through: matrix cracking in laminate, debonding between the fibre and matrix,  crack propagation and delamination, fibre debonding and pullout in the laminate and final rupture with extensive fibre pullout.

The article is systematically written with clear objectives and can be published in the present form. 

Author Response

Response to Reviewer 2 Comments

Point 1: This article entitled “Tensile Fracture Behavior and Characterization of Ceramic Matrix Composites” provides the example of the use of measurements involving an infrared camera and microstructural analysis using SEM pictures to explain the mechanical behaviour of ceramic matrix composites.

During this work, authors were able to confirm that the tensile failure of tested ceramic matrix composites runs through: matrix cracking in laminate, debonding between the fibre and matrix, crack propagation and delamination, fibre debonding and pullout in the laminate and final rupture with extensive fibre pullout.

The article is systematically written with clear objectives and can be published in the present form.

Response 1: Thank you very much for your most kind comments.

Reviewer 3 Report

I recommend a better description in figure 2. 0° Lamin and 90° Lamin  is not visible. Maybe a detail from these critical places would be proper as well as. 

I recommend a better description in Figure 3. What is what? Where is a camera, where is the sample ....

Author Response

Response to Reviewer 3 Comments

Point 1: I recommend a better description in figure 2. 0° Lamin and 90° Lamin is not visible. Maybe a detail from these critical places would be proper as well as.

Response 1: As the reviewer pointed out, the notation in Figure 2 was not clear. Therefore, in Figure 2, 0 and 90 degree Lamina were modified so that they appear well.

Point 2: I recommend a better description in Figure 3. What is what? Where is a camera, where is the sample ....

Response 2: As the reviewer pointed out, Figure 3 has been modified. High-speed infrared camera, testing grips, and CMC sample were added as captions.
